# Effect of Pesticides and a Long-Life Inoculant on Nodulation Process and Soybean Seed Quality during Storage

Rodrigo S. Araújo [1], Gisele C. Silva [1], Itamar R. Teixeira [1,*], Guilherme R. Silva [1], Bruna Mayara F. Souza [1], Ivano A. Devilla [1], Marcos Eduardo V. Araújo [2] and Paulo C. Corrêa [2]

[1] Anápolis Campus, Institute of Agricultural Sciences, Agricultural Engineering Course, State University of Goiás, Anápolis 75132-903, Goiás, Brazil; rodrigo.starneck@gmail.com (R.S.A.); gisele.carneiro@ueg.br (G.C.S.); guilhermesilvaromao@outlook.com (G.R.S.)

[2] Center of Agrarian Sciences, Viçosa Campus, Department of Agricultural Engineering, Federal University of Viçosa, Viçosa 36570-900, Minas Gerais, Brazil; copace@ufv.br (P.C.C.)

* Correspondence: itamar.texeira@ueg.br

**Abstract:** Inoculants containing strains of bacteria that fix atmospheric N that are tolerant or resistant to chemical products commonly used in seed treatment are classified as long-life inoculants, which contribute to improving the efficiency of nodulating bacteria in soybean crops. The objective of this work was to evaluate the effects of applying commonly used pesticides in seed treatment and a long-life inoculant, alone or in combination, on the nodulation process and seed quality of soybeans during storage. The first experiment was carried out in a greenhouse using a completely randomized design, in an 8 × 2 factorial arrangement, with three replications. The treatments consisted of applications of industrial seed treatments: control, MaximAdvanced, Fortenza, long-life inoculant, MaximAdvanced + Fortenza, MaximAdvanced + long-life inoculant, Fortenza + long-life inoculant, and MaximAdvanced + Fortenza + long-life inoculant. The seeds were sown in pots containing soils from two crop areas. The second experiment was carried out in a laboratory, using a completely randomized design in an 8 × 7 factorial arrangement, with four replications. The treatments consisted of the same eight treatments used in the greenhouse experiment, which were applied before packaging and storing the seeds for two months. The industrial seed treatment with the mixture of fungicide, insecticide, and inoculants improved the soybean photosynthesis and nodulation processes, regardless of the history of the area. Soybean seeds can be stored for commercial purposes for up to 40 days, regardless of the seed treatment with fungicide, insecticide, and long-life inoculant applied alone or in combination.

**Keywords:** *Glycine max* L.; industrial seed treatment; biological nitrogen fixation; nodulation efficiency; quality seed

## 1. Introduction

Soybean crops are currently widespread in the world due to their diverse uses in different sectors and by their significant presence in international markets, as they are a commodity. Brazil, followed by the US, is currently the world's largest oilseed-producing country [1], and soybeans are one of the main agricultural products exported by these countries.

Considering technological advancements in soybean crops, seed companies are adopting new techniques to maximize crop yields, for instance, the development of an industrial seed treatment process [2,3] called industrial seed treatment, which involves treating seed lots during the processing phase, followed by bagging and storage until sowing [4–10]. This innovative technique can be conducted using novel formulations containing fungicides, insecticides, nematicides, stimulants, micronutrients, and inoculants in a single treatment [11–13], enhancing the efficiency of the products, contributing to the protection of applicators, and preventing environmental contamination.

Early treatment may reduce the seed physiological quality of seeds during storage due to the potential phytotoxic effects that some active ingredients in the products may cause on the seeds. These phytotoxic effects can reduce seed germination and vigor, as well as seedling emergence, hindering the establishment of soybean plants and crop yield [14].

Most toxic effects to seeds are caused by treatments involving insecticides applied alone or combined with other chemicals, such as fungicide, micronutrients, and bio-stimulants. In [15], the authors evaluated a mixture of insecticide (thiamethoxam) with a complex of macro- and micronutrients (DimicronTMSp), and a bio-stimulant containing indolbutyric acid, kinetin, and gibberellic acid (Stimulate®); they found that the treatment with the nutrient complex presented the best results in the initial performance of soybean seeds, followed by the treatment with plant growth regulator with a bio-stimulating effect.

Insecticides have a negative effect on seed germination and seedling development. A study [8] showed that the use of the fungicide Derosal Plus® and the insecticide Cruiser, alone or combined, hindered seed quality and vigor of three soybean cultivars (M7110 IPRO, RR-8473RSF, and M7739 IPRO); the seeds were stored after chemical treatment and maintained commercial standards (>80%) until 60 days.

Seed treatments containing insecticides, fungicides, micronutrients, and bio-stimulants that can also interact with nitrogen-fixing bacteria can affect nodulation and photosynthesis processes and, consequently, grain yield; however, this effect depends on the interval between seed treatment and sowing. In [14], the authors found a negative effect on the nodulation process of soybean plants and consequently on grain yield, mainly when seeds were inoculated and treated with thiabendazole + fludioxonil + metalaxyl-M 30 days before sowing compared to seed inoculation and treatment 3 h before sowing.

Inoculants are widely used to supply N to soybean crops, mainly in Brazil and Argentina. Modern products, known as long-life inoculants, have recently been released on the market; they present high efficiency in promoting plant nodulation and can be used in combination with application of fungicides and insecticides before planting through industrial seed treatment [16]. These products include Granouro® and HicoatS30® from BASF, which can be applied to seeds up to 45 days before sowing, provided they are stored under proper conditions. However, research on this subject is still incipient.

Understanding factors that contribute to the variability in soybean responses to inoculation with long-life rhizobia and their mixtures with agricultural pesticides, such as fungicides and insecticides, applied through industrial seed treatment is an important step toward the development of sustainable agricultural systems while promoting increases in soybean production and reduction in costs [17]. Furthermore, there is a need for investigative studies on the dynamics of soybean seeds subjected to industrial seed treatments containing mixtures of insecticides, fungicides, and inoculants, followed by storage, as well as on the effects of these products on the nodulation process and photosynthesis efficiency of soybean plants.

In this context, the objective of this work was to evaluate the effects of applying commonly used pesticides in seed treatment and a long-life inoculant, whether alone or in combination, on the plant nodulation process and the quality of soybean seeds stored for two months.

## 2. Material and Methods

### 2.1. General Information

The first experiment was conducted in a greenhouse to assess the viability of chemical pesticides and inoculants, applied alone or in combination through industrial seed treatments, regarding soybean photosynthesis (chlorophyll index) and nodulation, using soils collected in areas with different soybean crop yields. The second experiment was conducted in a laboratory, involving evaluations of the same treatments investigated in the greenhouse experiment, focusing on their effects on the physiological quality of soybean seeds during storage.

## 2.2. Greenhouse Experiment

A completely randomized design was used, in an $8 \times 2$ factorial arrangement, with three replications. The treatments consisted of eight combinations of products for application through industrial seed treatment, involving commonly used pesticides in seed treatment (fungicides and insecticides) and a long-life inoculant, whether alone or in combination: control (T1); fungicide MaximAdvanced (T2); insecticide Fortenza (T3); long-life inoculant Nitragin CT 200 (T4); fungicide and insecticide (MaximAdvanced + Fortenza) (T5); fungicide and long-life inoculant (MaximAdvanced + Nitragin CT 200) (T6); insecticide and long-life inoculant (Fortenza + Nitragin CT 200) (T7); and fungicide, insecticide, and long-life inoculant (MaximAdvanced + Fortenza + Nitragin CT 200 (T8). Soybean seeds were sown in pots containing soils from two crop areas, one in the first year of growing soybeans and the other having been used for soybeans for several years.

The following products and rates were used for the seed treatments: Fortenza (insecticide) at 60 mL 100 kg$^{-1}$; Polymer (adhesive–insecticide)at 100 mL 100 kg$^{-1}$; MaximAdvanced (fungicide) at 100 mL 100 kg$^{-1}$; CRI (additive–fungicide) at 80 mL 100 kg$^{-1}$; Nitragin CTS 200—Semia 5079 and Semia 5080 (inoculant–peat, $7.0 \times 10^9$ cells gram) at 400 g 100 kg$^{-1}$; and Nitragin Power 200 (additive–inoculant) at 350 mL 100 kg$^{-1}$.

The soybean seeds were treated and immediately sown in 5L pots. Five seeds of the soybean cultivar 73,170 RSE IPRO were sown; the seedlings were thinned 10 days after emergence (DAE), leaving two plants per pot. Irrigation was carried out according to the crop's water needs.

The analyses were performed at the V5 stage (full flowering) by sampling the two plants in the pot with presence of shoots and roots. The root system was washed in running water to remove soil and aerial partswhen necessary. The plants were then sectioned at the stem base to evaluate the shoot and root systemsseparately.

The following plant morphophysiological characteristics were evaluated: chlorophyll content, evaluated using a portable chlorofiLOG (CFL 1030 m) on 10 fully expanded leaves from randomly selected mature branches at the middle part of each plant; numbers of viable, non-viable, and total nodules, evaluated using intact roots after separation and washing; all nodules found were removed, but only nodules with a diameter of 2 mm or more were evaluated; viability, determined by sectioning the nodules in half using a utility knife and identifying the pinkish coloration; dry weight of nodules, determined after counting and identifying the percentage of viable nodules, then samples were dried in an oven at 65 °C until constant weight to obtain the dry weight [18].

## 2.3. Lab Experiment

A completely randomized design was used, in an $8 \times 6$ factorial arrangement, with four replications. The treatments consisted of eight combinations of products for application through industrial seed treatment, using chemical pesticides (fungicides and insecticides) and a long-life inoculant, whether alone or in combination: control (T1), fungicide MaximAdvanced (T2), insecticide Fortenza (T3), long-life inoculant Nitragin CT 200 (T4), MaximAdvanced + Fortenza (T5), MaximAdvanced + long-life inoculant (T6), Fortenza + long-life inoculant (T7), and MaximAdvanced + Fortenza + long-life inoculant (T8). The seeds were stored for two months; evaluations were carried out every ten days (10, 20, 30, 40, 50, and 60 days). The tested treatments were the same as those described in the greenhouse experiment, including the rates.

The treated seeds were packed in raffia sacks and stored in a cold chamber, at a temperature of approximately 12 °C and a relative humidity of approximately 45%, to minimize metabolic activities and, consequently, loss of seed viability and vigor. The seeds were evaluated every 10 days, using the following tests:

- Moisture content: Determined by drying two 25-seedsubsamples per plot in a forced-air circulation oven at $105 \pm 3$ °C for 24 h, according to the Rules for Seed Analysis (RAS) [19].

- Germination: Evaluated using four 50-seed subsamples, arranged on germitest paper as substrate, which was moistened with distilled water corresponding to 2.5-fold the dry paper weight; 50 seeds (replications) were wrapped in the paper and placed in a germinator at a temperature of 25 ± 2 °C. Evaluations were conducted on the eighth day after the test started by calculating the percentages of normal and abnormal seedlings and dead seeds [19].
- First germination count: Conducted simultaneously with the germination test, with evaluations on the fifth day after the germination test started. The percentage of normal seedlings was calculated [19].
- Accelerated aging: 250 seeds (replications) were distributed on a metal screen fixed and suspended inside a plastic box (gerbox) containing 40 mL of water; the box was maintained at 41 °C and 100% relative humidity for 48 h in a germinator [20]. The seeds were then subjected to germination test, as previously described, and the percentage of normal seedlings was determined on the fifth day after the beginning of the test.

### 2.4. Statistical Analysis

The obtained data were subjected to analysis of variance ($p \leq 0.05$) and, when significant, Tukey's mean comparison test was applied for qualitative factors, whereas regression analysis was applied for quantitative factors. Statistical analyses were conducted using the software R 4.2.

### 3. Results

#### 3.1. Greenhouse Experiment: Chlorophyll Index and Nodulation of Plants

The result of the analysis of variance showed that all the evaluated variables were affected by the seed treatment factor. Contrastingly, the effects of factor interaction and the soil factor showed no significant differences for any of the evaluated variables. The chlorophyll index was significantly affected at 0.01 probability by the seed treatments with the different combinations of the evaluated products.

The treatments that presented the highest chlorophyll index were T8 (fungicide, insecticide, and inoculant) with 44.9, followed by T4 (inoculant only) with 44.3, which statistically differed from the others (Table 1). The other treatments resulted in higher chlorophyll contents than the control, however with no statistical difference.

**Table 1.** Mean chlorophyll index in leaves of soybean plants from seeds subjected to seed treatment.

| Treatments | Chlorophyll Index |
|:---:|:---:|
| T8 (fungicide + insecticide + inoculant) | 44.9 [1] ± 5.4 [2] A |
| T4 (inoculant) | 44.3 ± 4.7 A |
| T6 (fungicide + inoculant) | 42.5 ± 4.4 B |
| T7 (insecticide + inoculant) | 42.2 ± 5.0 B |
| T2 (fungicide) | 41.9 ± 5.1 B |
| T3 (insecticide) | 41.0 ± 4.8 B |
| T5 (fungicide + insecticide) | 40.0 ± 4.3 B |
| T1 (control) | 39.7 ± 4.1 B |
| MG [3] | 42.1 |

Means followed by the same letter in the columnsare not significantly different from each other by the Tukey's test at 0.05 probability; [1] mean; [2] SD = standard deviation; [3] MG = mean general.

No negative effect was found for the combination of these products regarding nodulation (Table 2). The highest results of total and viable nodules were found for the treatments T4 and T8. The overall mean dry weight of nodules was 0.80 g (Table 3). The lowest dry weights of nodules were found for treatments that included insecticide, fungicide, or insecticide + fungicide mixture.

**Table 2.** Mean percentages and standard deviation of viable nodules (VN), non-viable nodules (NVN), and total nodules (TN) in soybean plants from seeds subjected to seed treatment.

| Treatments | VN | Treatments | NVN | Treatments | TN |
|---|---|---|---|---|---|
| T4 | $52.8^{1} \pm 7.6^{2}$ A | T3 | $3.5 \pm 0.05$ A | T4 | $53.6 \pm 6.3$ A |
| T8 | $42.8 \pm 5.3$ B | T5 | $3.3 \pm 0.04$ A | T8 | $44.3 \pm 5.4$ A |
| T7 | $37.0 \pm 4.8$ C | T2 | $2.6 \pm 0.04$ A | T7 | $39.3 \pm 4.6$ B |
| T6 | $35.8 \pm 6.4$ C | T7 | $2.3 \pm 0.03$ A | T6 | $37.6 \pm 5.3$ B |
| T2 | $30.5 \pm 4.7$ C | T6 | $1.8 \pm 0.02$ B | T3 | $33.6 \pm 4.7$ B |
| T3 | $30.1 \pm 3.2$ C | T8 | $1.5 \pm 0.01$ B | T2 | $33.1 \pm 4.5$ B |
| T1 | $29.1 \pm 3.9$ C | T1 | $1.5 \pm 0.02$ B | T1 | $30.6 \pm 4.2$ B |
| T5 | $26.8 \pm 2.6$ C | T4 | $0.8 \pm 0.01$ B | T5 | $30.1 \pm 4.3$ B |
| MG [3] | 35.6 | - | 2.1 | - | 37.8 |

T1 = control; T2= fungicide; T3 = insecticide; T4 = inoculant; T5 = fungicide and insecticide; T6 = fungicide and inoculant; T7 = insecticide and inoculant; T8 = fungicide, insecticide, and inoculant. Means followed by the same letter in the columns are not significantly different from each other by the Tukey's test at 0.05 probability; [1] mean; [2] SD = standard deviation; [3] MG = mean general.

**Table 3.** Mean percentages of dry weight of nodules in soybean plants from seeds subjected to seed treatment.

| Treatments | Dry Weight of Nodules (g) |
|---|---|
| T4 (inoculant) | $1.22^{1} \pm 0.07^{2}$ A |
| T8 (fungicide + insecticide + inoculant) | $1.09 \pm 0.05$ A |
| T7 (insecticide + inoculant) | $0.94 \pm 0.05$A |
| T6 (fungicide + inoculant) | $0.89 \pm 0.04$ A |
| T3 (insecticide) | $0.66 \pm 0.04$ B |
| T2 (fungicide) | $0.57 \pm 0.04$ B |
| T5 (fungicide + insecticide) | $0.55 \pm 0.04$ B |
| T1 (Control) | $0.51 \pm 0.03$ B |
| MG [3] | 0.80 |

Means followed by the same letter in the columns are not significantly different from each other by the Tukey's test at 0.05 probability; [1] mean; [2] SD = standard deviation; [3] MG = mean general.

### 3.2. Lab Experiment: Seed Physiological Quality

According to the results of the analysis of variance, the interaction between factors (seed treatment and storage time) was significant for germination percentage. The evaluated soybean seeds exhibited differences in viability during storage based on applied treatment (Figure 1). Treatment T8 promoted a lower germination percentage than the other treatments, resulting in a mean of 81% after 60 days of storage.

Treatments T4 (inoculant only) and T1 (control) had the least effect on physiological quality throughout storage (Figure 2). Contrastingly, T8 was detrimental to seed vigor immediately after treatment application and throughout the entire storage period, presenting percentages below 70% in the first evaluation and reaching the lowest percentage at the end of storage (59%). These results are consistent with those obtained in the germination test.

Seeds treated with inoculant (T4) showed the highest germination percentage (69%), significantly differing from those in T1 (control), which presented the second-highest germination percentage (63%) at the end of storage; both statistically differed from the other treatments (Figure 3), confirming the results found in the germination and first count tests.

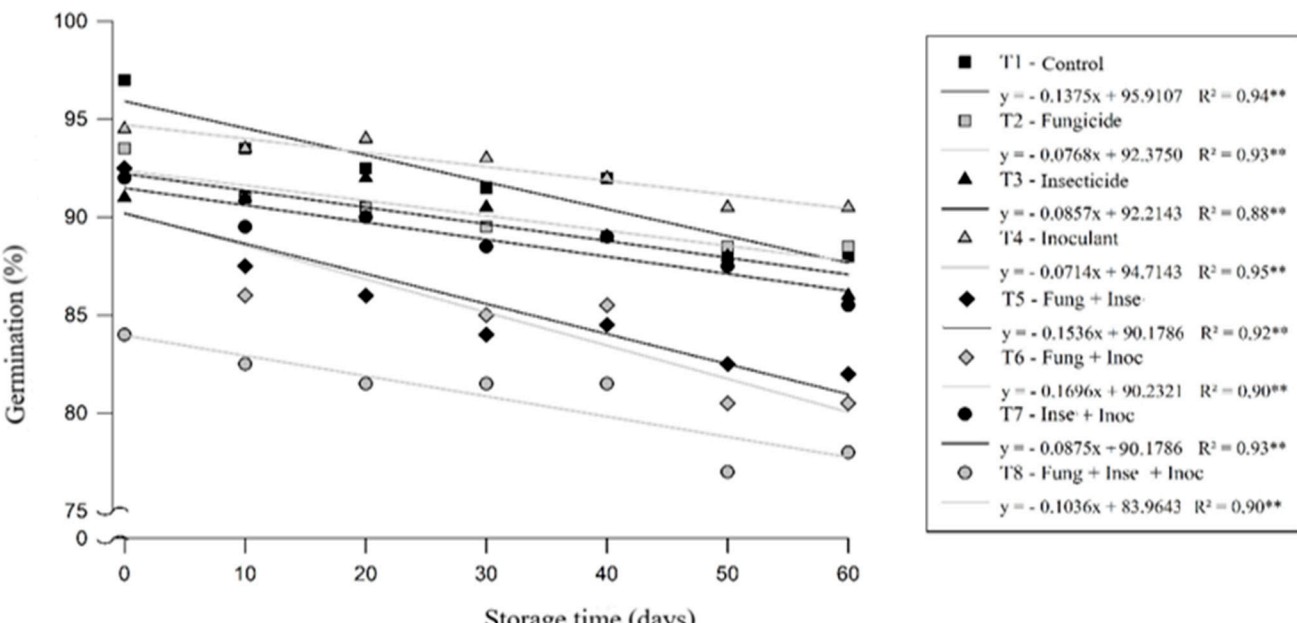

**Figure 1.** Germination percentage after different storage times of soybean seeds treated or not with pesticides and inoculant.

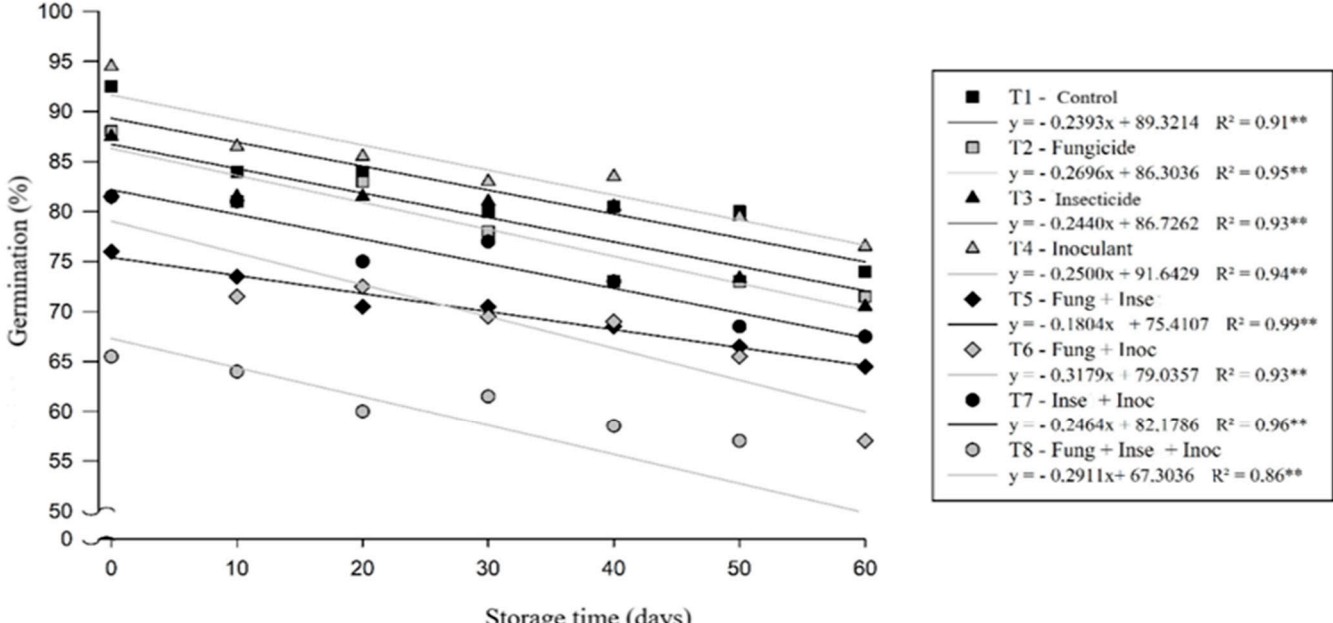

**Figure 2.** Germination percentage in the first count test after different storage times of soybean seeds treated or not with pesticides and inoculant.

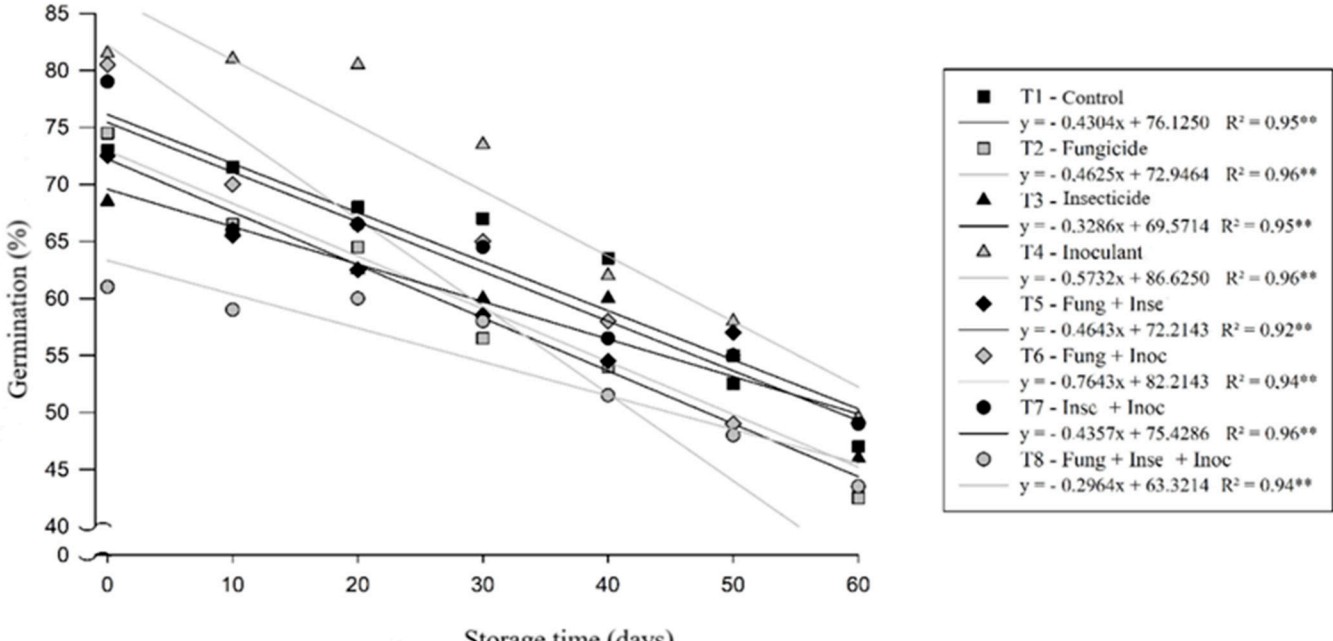

**Figure 3.** Germination percentage of accelerated aging test after different storage times of soybean seeds treated or not with pesticides and inoculant.

## 4. Discussion

An evaluation of different forms of application and rates of *Bradyrhizobium japonicum* to soybean crops developed by [21] showed significant effect of inoculant application, which promoted higher chlorophyll content compared to the control, as well as a mean chlorophyll index of 40.2. These results are consistent with those found in the present study, which found a mean chlorophyll index of 42.1 and the highest chlorophyll indices for treatments composed of mixtures that included the inoculant *B. japonicum*.

The mean chlorophyll index obtained by foliar application of different rates of *Bradyrhizobium* to soybean crops found by [22] was close to that found in the present study (38.9); however, the foliar application of inoculant resulted in a different chlorophyll index than those found for its application to seeds. They found that the control plants presented a more satisfactory chlorophyll index; thus, seed inoculation is more efficient than foliar application when evaluating chlorophyll index.

Chlorophyll is the main pigment responsible for capturing light energy in the photosynthesis process. Additionally, it is one of the main factors related to the photosynthetic efficiency of plants, directly affecting plant adaptability and growth [21].

The combinations of the evaluated products presented no negative effects on nodulation; a similar result was found by [22]. The highest numbers of total and viable nodules were found for plants in the treatments T4 and T8, denoting that the bacterial strain used is not susceptible to mixtures with pesticides, according to reports by [16].

The means found for viable and total nodules (37.7 and 37.8, respectively) are consistent with those found by [18], who evaluated the effect of different soybean seed treatments and found similar results, with a mean total nodulation of 42.1. The authors of [21] evaluated different inoculation rates and found similar means for nodulation; thus, the use of mixtures had no negative effects on soybean nodulation. Furthermore, [23] found that the number and weight of nodules obtained after inoculation with BR86 bacterial strain were not affected by insecticides; they reported that reductions in nodulation depend on the bacterial strain used. Thus, the strain used in the present study was not susceptible to mixtures with pesticides.

The highest results of viable and total nodules were obtained in the treatments T4 and T8, thus reinforcing that the bacterial strain used in the treatments is not susceptible to mixtures with pesticides, corroborating the results of [23].

The overall mean dry weight of nodules found (0.80 g) was higher than those reported by [18], who found a mean of 0.30 g when evaluating inoculant application methods and their effects on nodulation, as well as the addition of bioregulator to industrial treatments on the quality of soybean seeds. This difference can be attributed to the different evaluation timing; during the flowering stage, the nodules are at the peak of their development, thus presenting higher nodule weights.

The lowest dry weights of nodules were found for treatments including insecticide, fungicide, or a mixture of insecticide and fungicide. Similar results were found by [24], who found a 70% reduction in soybean plant biomass when adding fungicides (carboxin + thiram) and inoculant, compared to inoculation without fungicides. The authors of also reported decreases in shoot dry weight when seeds were treated with insecticides.

The industrial seed treatment containing a mixture of fungicide, insecticide, and long-life inoculant (MaximAdvanced + Fortenza + Nitragin CT 200) improved photosynthesis (chlorophyll index) and the nodulation process in soybean plants, regardless of the soybean crop area history. These results can be attributed to application of the industrial seed treatment two hours before sowing, which is close to the three-hour interval recommended by [14]. In this situation, there was a greater interaction of the inoculated products with the soil particles, mainly the fungicide and insecticide, thus confirming the higher effectiveness of these products in the control of diseases and pests in the soil, leading to improvements in nodulation and suitable conditions for photosynthesis.

The highest germination percentages were found at the beginning of storage in the treatments T4 (long-life inoculant) and T1 (control), which were higher than 92% and 90%, respectively, but with decreases in the number of normal plants at the time of the evaluations, as well as the other treatments.

The use of long-life inoculants stands out as a potential practice for soybean crops, as reported by [17], because the application of long-life inoculants through seed treatment is practical for growers and efficient in supplying N. Furthermore, the results indicate that under laboratory conditions, seed treatments with fungicides, insecticides, and inoculants, applied alone or in combination, significantly affect the initial seed development and induce physiological changes in seeds.

None of the tested treatments showed a seed germination mean lower than the minimum required standard for commercialization of soybean seeds, i.e., 80% [19], at the end of the storage period (60 days), even the treatment T8, which resulted in seeds with lower viability. This may be attributed to the direct contact of the products with the seeds during storage, which compromised the seed physiology and may explain the different results obtained for nodulation and photosynthesis from seeds subjected to the same treatments but to interaction of the applied products with the soil.

A study [25,26] showed higher proportion of normal soybean seedlings under absence of chemical treatments over a 45-day storage period; however, chemical treatments did not affect germination up to 40 days of storage, and the seeds presented germination percentages below the recommended levels for commercialization [19].

The evaluated soybean seeds differed in vigor during the storage period depending on the applied treatments. The highest percentages of normal seedlings were found at the beginning of storage and decreased over the subsequent evaluations. The highest percentages were found for seeds of the control treatment and differed significantly over the storage period, denoting that the seed treatment with chemicals hindered the seed vigor since the beginning of storage. In [4], the authors evaluated the use of insecticides on the quality of soybean seeds during storage and concluded that seed treatments with these products negatively affected germination and first germination count over a 30-day storage period, confirming the results found in the present study.

The accelerated aging test provided similar resultsto those of the previous evaluations, following the same dynamics. However, this test involves subjecting the seeds to stresses from humidity and temperature [27,28], followed by conducting a germination test with the aged seeds. This process explains the lower mean vigor of seedlings obtained in the accelerated aging test compared to those found in the germination and first count tests.

## 5. Conclusions

Treatments of soybean seeds with fungicide, insecticide, and inoculant, whether applied alone or in combination, improve photosynthesis (chlorophyll index) and the nodulation process in soybean plants grown in areas with a first-year or several-year history of soybean crops. The use of chemical pesticides (fungicide and insecticide) and their combinations with a long-life inoculants negatively affect the physiological quality of soybean seeds. Soybean seeds can be stored for a period of 40 days for commercial purposes, regardless of the seed treatment with fungicide, insecticide, and long-life inoculant products, whether applied alone or in combination.

**Author Contributions:** Conceptualization, R.S.A. and I.R.T.; methodology, R.S.A., G.C.S. and I.R.T.; investigation, R.S.A., B.M.F.S., G.R.S. and I.R.T.; formal analysis, R.S.A., B.M.F.S., G.R.S. and M.E.V.A.; funding acquisition, G.C.S. and I.R.T.; supervision, G.C.S. and I.R.T.; writing—original draft, R.S.A.; writing—review and editing, I.A.D., M.E.V.A. and P.C.C. All authors have read and agreed to the published version of the manuscript.

**Funding:** Graduate Program Master in Agricultural Engineering—UEG; Financial Resource from Call No. 21/2022; Term of Commitment No. 000036040537; Process SEI No. 202200020020855.

**Data Availability Statement:** Data are available upon request to the authors.

**Acknowledgments:** To CNPq for granting the research productivity grant, process 313501/2021-1, to the third author.

**Conflicts of Interest:** The authors declare no conflict of interest.

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
