# Peer review of "Effect of Pesticides and a Long-Life Inoculant on Nodulation Process and Soybean Seed Quality during Storage"

_agronomy, doi:10.3390/agronomy13092322_

Round 1

Reviewer 1 Report

Dear authors,

I found your article very interesting and well written with minor changes to be make. Please, consider the following comments:

-          Lines 1 to 4: The title is not grammatically correct. I suggest to change the first word (Efficiency) by “Effect” or “Impact”.

-          Line 14: N2 needs to be subscripted.

-          Lines 24 to 26: To which extent are the following abbreviations needed in the abstract? “(T1–control; T2–MaximAdvanced; T3–Fortenza; T4–long-life inoculant;T5–MaximAd-24 vanced + Fortenza); T6–MaximAdvanced + long-life inoculant; T7–Fortenza + long-life inoculant; 25 T8–MaximAdvanced + Fortenza + long-life inoculant)”. Can they be omitted without affecting the message of the abstract? If so, please remove them.

-          Line 35: Maybe better to write as key word: “seeds’ industrial treatment” or as you are defining in your article “industrial seed treatment”.

-          Line 75: BASF with capital letters.

-          Line 82 to 85: The introduction was good as it provides with sufficient background for the investigation. However, I suggest to include at least some information on the relevance of the process of nodulation, to avoid reaching the last paragraph of the introduction to mention this plant process.

-          Line 164: Either 5th or “fifth”.

-          Line 167: Include spaces before and after ≤.

-          How is possible that there is no direct relation between the nodulation and the germination? For instance, T8 promoted nodulation but prevented germination.

-          Lines 1 to 4: The title is not grammatically correct. I suggest to change the first word (Efficiency) by “Effect” or “Impact”.

-          Line 14: N2 needs to be subscripted.

-          Lines 24 to 26: To which extent are the following abbreviations needed in the abstract? “(T1–control; T2–MaximAdvanced; T3–Fortenza; T4–long-life inoculant;T5–MaximAd-24 vanced + Fortenza); T6–MaximAdvanced + long-life inoculant; T7–Fortenza + long-life inoculant; 25 T8–MaximAdvanced + Fortenza + long-life inoculant)”. Can they be omitted without affecting the message of the abstract? If so, please remove them.

-          Line 35: Maybe better to write as key word: “seeds’ industrial treatment” or as you are defining in your article “industrial seed treatment”.

-          Line 75: BASF with capital letters.

-         Line 164: Either 5th or “fifth”.

-          Line 167: Include spaces before and after ≤.

Author Response

Dear Reviewer, the following changes to the manuscript were made:

- English language review.

- Lines 1 to 4: Word "efficiency" replaced by "effect".

- Line 14: "N2" has been overwritten.

- Lines 24 to 26: the abbreviations of the treatments were removed.

- Line 35: The term “industrial seed treatment” was used in the keywords.

- Line 75: The word "BASF" is capitalized.

- Line 82 to 85: A paragraph about the nodulation process was included.

- Line 164: The term "fifth" was used.

- Line 167: spaces were included "(p ≤ 0.05)".

- An explanation was provided in the Results for the fact that treatment 8 (T8) has contributed to improved plant nodulation, and on the other hand, impaired seed germination.

We appreciate the suggestions!

PhD. Itamar Teixeira

Reviewer 2 Report

Agronomy-2543582 is an extremely simple work. It provides the low information to the researchers and readers. Obviously, its workload cannot meet the publishing requirements of Agronomy. Thus, I suggest that it be published in the form of a short communication after major modification.

1. Manuscript is an extremely simple work, its workload obviously cannot meet the publishing requirements of Agronomy.

2. The description of the methods (Line 19-30) in the abstract should be heavily abbreviated, and what kind of results and conclusions are elaborated.

3. Manuscript must be through language editing.

4.The logic and neat of introduction need to be further improved, some paragraphs long and some short are messy.

5. The concentrations of treatments were not introduced. And what is the basis for their use of this treatments? This is the biggest problem.

6. There should be spaces between numbers and letters (such as line 107-110 etc.), please check the full text.

7. All data in Table 1-3 should be supplemented by standard deviation for evaluating the reliability of data.

8. Line 167: ‘P’ should be italic. Please check the full text.

9. Results and discussions should be written separately.

Manuscript must be through language editing.

Author Response

Dear reviewer, thank you for your suggestions. Here are comments on the suggestions:

  1. The manuscript generated information on the effect of using recently released chemicals on the world market, with potential use associated with long-life inoculants, also a new product on the market, on the nodulation processes of plants after emergence and the quality of seeds during storage, submitted to industrial seed treatment. It should be noted that the chemical products and the inoculant used, as well as the use of industrial seed treatment are relatively new technologies worldwide, therefore there is little information in the literature about their potential use in soybean crop. Give the relevance of the subject!
  2. Line 19-30 - In the Abstract, abbreviations were removed and conclusions expanded.
  3. The manuscript was revised in English
  4. The wording of Introduction has been improved making it more objective. The size of the paragraphs has been standardized. Added a paragraph on the plant nodulation process.
  5. The concentrations of chemical products were already mentioned in the methodology, lacking only the inoculant, which was added.
  6. Added spaces between numbers and letters throughout the text
  7. It was added due to default in the three tables.
  8. ‘P’ has been italicized
  9. The Results and Discussion items were separated

Round 2

Reviewer 2 Report

The manuscript was well improved and can be accepted for publication.

The manuscript was well improved and can be accepted for publication.